# Thiol-Functional Polymer Nanoparticles via Aerosol Photopolymerization

**DOI:** 10.3390/polym13244363

**Published:** 2021-12-13

**Authors:** Narmin Suvarli, Iris Perner-Nochta, Jürgen Hubbuch, Michael Wörner

**Affiliations:** Institute of Process Engineering in Life Science, Section IV: Biomolecular Separation Engineering, Karlsruhe Institute of Technology (KIT), 76131 Karlsruhe, Germany; narmin.suvarli@kit.edu (N.S.); iris.perner.nochta@kit.edu (I.P.-N.); juergen.hubbuch@kit.edu (J.H.)

**Keywords:** thiol-ene polymerization, aerosol photopolymerization, polymer nanoparticles, thiol-functional nanoparticles

## Abstract

Spherical, individual polymer nanoparticles with functional –SH groups were synthesized via aerosol photopolymerization (APP) employing radically initiated thiol-ene chemistry. A series of various thiol and alkene monomer combinations were investigated based on di-, tri-, and tetrafunctional thiols with difunctional allyl and vinyl ethers, and di- and trifunctional acrylates. Only thiol and alkene monomer combinations able to build cross-linked poly(thio-ether) networks were compatible with APP, which requires fast polymerization of the generated droplet aerosol during the photoreactor passage within a residence time of half-minute. Higher monomer functionalities and equal overall stoichiometry of functional groups resulted in the best nanoparticles being spherical and individual, proven by scanning electron microscopy (SEM). The presence of reactive –SH groups in the synthesized nanoparticles as a basis for post-polymerization modifications was verified by Ellman’s test.

## 1. Introduction

Polymer nanomaterials have found applications in many fields of science and technology—e.g., drug delivery [1,2], tissue engineering [3], biophotonics [4,5], water treatment [6,7], food processing [8,9]. The synthesis of polymer nanoparticles via liquid routes can be achieved using either preformed polymers or by polymerization of monomers [10]. Concerning preformed polymers, several techniques have been reported. Among these, solvent evaporation is widely used, in particular for the formation of nanoparticles for drug delivery applications [11,12]. A drawback of this technique is the requirement for high amounts of hazardous organic solvents. Other methods that use preformed polymers—such as dialysis [13], nanoprecipitation [14], and salting-out [15]—usually do not involve the use of hazardous materials. Polymerization of monomers in aqueous dispersed reaction systems—such as conventional emulsion [16], miniemulsion [17], microemulsion [18] polymerizations—result in polymer micro- and nanoparticles. These methods are used when a polymer is designed for a specific range of applications [10]. Emulsion polymerization is one of the most often employed techniques in industry, where the use of surfactants is required and mostly thermal initiation is applied, consuming energy for the emulsification process as well. Different emulsification procedures can be also combined with photochemical techniques, where photons are used to initiate the polymerization reaction, in order to produce polymer particles. Photopolymerization of multifunctional monomers is an effective method to produce cross-linked polymers/coa-tings within a very short time and has been applied, for example, in curing technologies [19].

Aerosol-based polymerization methods can be applied for the synthesis of nanomaterials as an alternative to liquid-based processes. Integrated and continuous gas-phase processes synthesize nanostructures of higher purity compared to the liquid-phase methods. Nanoparticles can be produced via either evaporation and crystallization of atomized droplets, or condensation and coagulation from nucleation and growth of gas particles [20]. Microscale polymer particles (5–100 µm) have been produced via atomization of monomer solutions with a vibrating-orifice aerosol generator and subsequent photopolymerization to accomplish droplet-to-particle conversion with UV-irradiation [21,22]. To overcome the drawbacks of low droplet generation rate and size limitations within the µm-scale, a modified technique using a collision-type aerosol device was established which permits the generation of highly concentrated monomer droplet aerosols (>10^8^ droplets/cm^3^) in the nano-size domain (0.1–0.5 µm) [23]. The produced droplet aerosol is exposed to UV-irradiation in a flow-through reactor, where the droplets polymerize in a continuous process. In contrast to other methods that produce polymer nanoparticles, aerosol photopolymerization (APP) does not require the presence of surfactants, or the use of heating and hazardous solvents, resulting in highly pure polymer particles. The residence time of the aerosol inside the photoreactor is short, which requires fast polymerization kinetics that can be accomplished by various combinations of initiation and propagation rates [24]. For monomers with low propagation rates, APP is not a suitable method, but the addition of a small amount of an appropriate cross-linker can lead to faster polymerization and APP can be applied [23]. Various types of nanomaterials can be produced via APP: organic-inorganic hybrid nanoparticles [25], in-situ drug-loaded polymer nanocarriers [26], porous spherical nanoparticles, and nanocapsules [27,28].

Research in the field of thiol-ene chemistry has widened in recent years due to numerous benefits of the method: it is an easy, straightforward reaction that results in biocompatible materials [29,30,31,32,33,34]. The highly efficient thiol-ene ‘click’ reaction is one of the well-known methods in biochemistry [35]. Thiol-ene photopolymerization is a rapid process [36] uninhibited by oxygen that does not require the use of solvents [37] and exhibits delayed gelation [38]. The step-growth mechanism of the thiol-ene photopolymerization reaction is based on alternating propagation and chain-transfer steps [37,39]. Studies of thiol-ene photopolymerization reaction systems in solutions have shown that changes in the rate of thiol and -ene functional group concentrations depend on the ratio of kinetic constants of propagation and chain transfer [32]. When the chain-transfer is the limiting step, the polymerization kinetics of the thiol-ene photopolymerization do not depend on the concentration of -ene functional groups, it is rather first-order kinetics depending on the concentration of thiol groups, and, vice versa, when the propagation is the rate-limiting step [32,39]. Combination of thiol-ene chemistry with photochemistry in miniemulsion polymerization process produces semicrystalline poly (thioether ester) latex nanoparticles. This simple and environmentally friendly route utilizes efficient thiylradical addition on unsaturated monomers ensuring fast polymerization at room temperature that can be applied in continuous photoreactors [40].

The use of thiol-ene photopolymerization in combination with miniemulsification that produces polymer nanoparticles was reported [34,41,42,43]. To date, no application of the APP technique with thiol-ene chemistry was reported. In this paper, we present aerosol thiol-ene photopolymerization as a method to produce cross-linked, narrow-size, spherical individual polymer nanoparticles with active –SH groups. We studied a variety of thiol and alkene monomers to discover the most suitable set of combinations. Regarding future applications of the particles—e.g., by biofunctionalization—the availability of –SH groups was determined.

## 2. Materials and Methods

### 2.1. Chemicals

The following thiols and alkenes were used as monomers (Figure 1): diallyl adipate (DAA, TCI chemicals, Tokyo, Japan), tri(ethylene glycol) divinyl ether (TEG-DVE, Merck Group, Darmstadt, Germany), trimethylolpropane triacrylate (TMPTA, Sigma-Aldrich, St. Louis, MO, USA, contains 600 ppm monomethyl ether hydroquinone as inhibitor), neopentyl glycol diacrylate (NPG, Sigma-Aldrich, St. Louis, MO, USA), 1,3,5-triallyl-1,3,5-triazin-2,4,6-(1H,3H,5H)-trion (TATT, Sigma-Aldrich, St. Louis, MO, USA), trimethylolpropane tris (3-mercaptopropionate) (trithiol, Sigma-Aldrich, St. Louis, MO, USA 95%), tris(2-(3-mercaptopropionyloxy)ethyl) isocyanurate (TMPIC, Bruno Bock, Marschacht, Germany), ethylenbis(3-mercaptopropionat) (Dithiol, Bruno Bock, Marschacht, Germany), pentaerythrit-tetrakis-(3-mercapto-propionat) (Tetrathiol, Bruno Bock, Marschacht, Germany). 2-methyl-4′-(methylthio)-2 morpholinopropiophenone (MMP) purchased from Sigma-Aldrich (Merck Group, Darmstadt, Germany) was used as a photoinitiator. Ellman’s test was performed with 5,5′-dithiobis(2-nitrobenzoic acid) (DTNB, ReagentPlus^®^, 99%) purchased from Sigma Aldrich (Merck Group, Darmstadt, Germany).

### 2.2. Aerosol Photopolymerization Setup

The aerosol photopolymerization setup (Figure 2) consists of a collision-type atomizer (TOPAS^®^ ATM220), a photoreactor and a collection unit. The spray solution (1) placed in the aerosol generator is atomized with the stream of nitrogen inside the nozzle (2) forming a droplet aerosol. Droplets pass into the UV-irradiated reaction chamber (3), which consists of five 14-mm inner diameter fluorinated ethylene propylene (FEP) tubes connected to form a cycle inside the reactor. UV irradiation is provided by two irradiation devices, each equipped with three UV-fluorescent tubes (T-15.C, Vilber Lourmat, *λ_max_* = 312 nm–incident irradiance E = 15.4 mW/cm^2^ (measured via UV-Pad-E from Opsytec Dr. Gröbel GmbH)) (4) facing the reactor. Polymerized nanoparticles are collected inside a BOLA^®^ flow filtration housing on 0.1 µm hydrophobic PVDF membrane filters (Durapore^®^) (5). The nanoparticles are then left to dry on the filter overnight. An aerosol photopolymerization reaction can yield between 50 and 200 mg of nanoparticles per hour of operation, depending on the concentration of monomers in the spray solution and pressure of nitrogen. At 1 bar (1.28 L/min nitrogen flow rate) and 0.37 L volume of FEP tubes an approximate residence time can be calculated to be 28 s.

### 2.3. Experiments

#### 2.3.1. Preparation of Spray Solutions

The formulations of spray solutions are presented in Table 1. The spray solution for the aerosol photopolymerization process was prepared by mixing the thiol and alkene monomers and then adding the solvent and photoinitiator in an amber flask wrapped with aluminum foil to prevent premature polymerization. In some cases, thiol and alkene monomers were not miscible, but with the addition of an appropriate solvent have formed a homogeneous solution. The solvent of choice was ethanol, however as Tetrathiol and TMPIC were not miscible with ethanol, the atomization was carried out in acetone (AcO) or acetonitrile (MeCN). The photoinitiator was added to all spray solution formulations and stirred for five minutes before the atomization; the quantity of photoinitiator in all spray solutions corresponded to 1 wt% of the quantity of two monomers combined. Table 1 contains some of the formulations of the spray solutions prepared for aerosol photopolymerization (in addition, see Appendix A).

#### 2.3.2. Ellman’s Tests

The qualitative analysis of reactive –SH groups of the polymer nanoparticles was carried out using Ellman’s reagent–DTNB. 4 mg of DTNB was dissolved in 5 mL of phosphate buffer saline (PBS) pH = 7.4. 5 mg of nanoparticles were dispersed in 5 mL of PBS pH = 7.4 and sonicated in an ultrasonic bath (Sonorex Digital 10 P) at 80% amplitude for 10 min at room temperature. The dispersions of nanoparticles (100 µL) from different samples (P1–P4, P9–P13, P15–P19, Table 1) were placed inside the Thermo Scientific™ 96-Well-Microtiter plates, 100 µL of DTNB solution was introduced to the dispersion and a coloration change was observed.

### 2.4. Analysis Methods

#### 2.4.1. Scanning Electron Microscopy (SEM)

Nanoparticles were dispersed in acetone and 100 µL of the dispersion was distributed on silicon wafers, dried, and sputtered with platinum. The plate was then placed inside LEO1530 (Carl Zeiss AG, Oberkochen, Germany) microscope. The SEM images of nanoparticles were taken at three different magnifications—2000×; 10,000×; and 25,000×.

#### 2.4.2. Image Analysis of SEM Micrographs

SEM images of some samples were analyzed for size distribution studies using open-source ImageJ software. The diameter of nanoparticles was determined manually for each nanoparticle on the micrograph, and the count rate of nanoparticles was equalized for each sample. The histograms of the determined diameters were built using OriginLab^®^ software.

## 3. Results and Discussion

### 3.1. Polymer Nanoparticles from Aerosol Thiol-Ene Photopolymerization

Some acrylate and vinyl ether monomers have already been used to produce polymer nanoparticles via APP [23]. In this research thiol and alkene monomers are tested employing aerosol photopolymerization. A great variety of thiol and alkene monomers with different quantities of functional groups (–SH and –C=C–, respectively) were studied to determine the set of monomer combinations suitable for the synthesis of spherical, individual poly(thio-ether) nanoparticles. For each alkene monomer, homopolymerization was attempted before investigating the thiol-ene photopolymerization. Not all used alkene monomers formed polymers in a radically initiated aerosol photopolymerization process. Due to the short residence time inside the photoreactor, only fast propagating monomers can form homopolymer nanoparticles via APP. The extremely low reactivity of allyl and vinyl ether towards free radical polymerization is most probably a consequence of the high stability of the propagating radicals. Nevertheless, cationic chain-growth reaction can be very fast and has shown great compatibility with APP using TEG-DVE as a monomer [44]. Allyls and vinyl ethers cannot be homopolymerized employing free radical initiation because of degrative chain transfer leading to oligomers or low-molecular-weight products. Only multifunctional monomers (as for TATT) and high-initiator-concentration polymeric products had been obtained after long reaction times [45].

Acrylate monomers, on the other hand, react in a free-radical chain-growth polymerization [46] fast enough to proceed under APP conditions. The products of these reactions can be analyzed via SEM (Appendix A).

The aerosol photopolymerization of spray solutions specified in Table 1 leads to the formation of polymer nanoparticles; SEM images of these nanoparticle samples are presented in Figure 3. Polymer nanoparticles produced from monomers with a 1:1 stoichiometric ratio of functional groups are presented in this figure. Other functional group ratios (especially with a higher amount of thiol) showed agglomerated or highly agglomerated polymer material (Appendix A). Additionally, it has to be mentioned that sample preparation for SEM imaging might result in increased agglomeration of polymer nanoparticles.

The polymer nanoparticles produced from the combination of trithiol with TMPTA (triacrylate) (P1) appear less agglomerated than combinations of trithiol with other alkene monomers (DAA, NPG, TEG-DVE) in ethanol. The combination of TMPIC with TMPTA (P6) and Tetrathiol with TMPTA (P11) also produce individual polymer nanoparticles. Solid-state ^13^C NMR spectra of TMPTA (M1) homopolymer and trithiol-TMPTA (P1) he-teropolymer nanoparticles (Appendix A) reveal that in the P1 nanoparticles all TMPTA double bonds reacted, whereas in the TMPTA-homopolymerization reaction the conversion of the double bonds was lower. The results obtained from ^13^C NMR were confirmed with Fourier-transform infrared (FT-IR) analysis (Appendix A). In the case of P1, both homo- (acrylic chain) and heteropolymerization (thiol-ene step-growth) can be competing reactions. A mixed step and chain-growth polymerization reaction has been observed [35,47] for acrylic monomers in thiol-ene photopolymerization reaction systems. The so-called mixed-mode polymerization is based on free radical polymerization reactions that show different development of molecular weight compared to chain and step polymerization individually, thus resulting in a rapid polymerization reaction with cross-linking (in case of multifunctional monomers) [48].

A similar phenomenon can be observed for a combination of NPG (diacrylate) and thiols. Although NPG homopolymerization (M2, Appendix A) reaction produces individual polymer nanoparticles, thiol-ene heteropolymerization with trithiol (P2) resulted in highly agglomerated polymer material. The behavior of other thiol monomers with NPG in P12 (Tetrathiol) results in a low degree of agglomeration and P7 (TMPIC) results in the formation of highly agglomerated material. These outcomes also showcase the possibility of occurrence of mixed-mode polymerization, but one can expect that the degree of cross-linking is lower than in the case of TMPTA because monomers with three or more functional groups are more prone to the formation of highly cross-linked polymers [38]. This can relate to the formation of more individual nanoparticles in P1 (trithiol + TMPTA) formulation, compared to P2 (trithiol + NPG). A combination of trifunctional thiol with trifunctional alkene (3FG-3FG) is expected to result in higher cross-linking within the polymer matrix, compared to combinations of trifunctional thiol with difunctional alkene (3FG-2FG), providing the stoichiometry of functional groups is kept equal.

Polymer nanoparticles can be produced from combinations of DAA (diallyl) with tri- and tetrafunctional thiols (P3, P8, P13) and TEG-DVE (divinyl ether) with the same set of thiols (P4, P9, P14). In the case of TMPIC (P8, P9), the nanoparticles appear to have a high degree of agglomeration compared to other thiols; when using another solvent (MeCN) in samples P17 and P18 (Appendix A), the nanoparticles still appear heavily agglomerated. In the case of combinations of DAA (P3, P8, P13) and TEG-DVE (P4, P9, P14), the mixed-mode can be excluded, because almost no homopolymerization of alkene is expected to take place. Therefore, the products of P3, P4, P8, P9, P13, P14 formulations are products of the thiol-ene polymerization reaction, exclusively. As in the case of NPG, DAA and TEG-DVE each possess only two functional groups, which may lead to a lower degree of cross-linking in thiol-ene reactions with tri- and tetrafunctional thiols, compared to reactions with the trifunctional monomer TMPTA.

With the set-up used, TATT (triallyl) cannot yield homopolymers via free-radical aerosol photopolymerization; however, spherical and individual polymer nanoparticles can be produced via thiol-ene photopolymerization reactions (P5, P10, P15) with tri- and tetrafunctional thiols. In the case of these reactions, only step-growth thiol-ene polymerization has to be taken into account, and highly cross-linked poly(thio-ether) nanoparticles are expected to be produced considering that TATT comprises three -ene functional groups.

Aerosol photopolymerization of alkenes with a difunctional thiol (dithiol) was also studied and the results are presented in Appendix A.

The results presented in this section underline the use of thiol-ene chemistry for aerosol photopolymerization and showcase a set of monomer combinations that results in spherical individual polymer nanoparticles. Trifunctional thiols with trifunctional alkene monomers in an equal stoichiometry of functional groups are the best tool in producing cross-linked and individual polymer nanoparticles.

### 3.2. Ellman’s Test

The presence of –SH groups within the polymer particles can be determined using FTIR; however, the –SH stretching mode shows a very weak absorption peak in trithiol. In Appendix A, the signals of –SH groups are visible for P1 polymer nanoparticles, which were synthesized through a mixed mode polymerization mechanism. In the case of the pure thiol-ene polymerization (P5), no trace of –SH groups is observed. However, a more precise method is necessary due to the low sensitivity of ATR-FTIR towards detecting thiol groups.

Elman’s test was carried out with polymer nanoparticles (from Table 1) produced via aerosol thiol-ene photopolymerization to determine the presence of reactive –SH groups. Homopolymers of TMPTA and NPG (M1 and M2, respectively) do not show coloration, as expected, whereas all polymer nanoparticles produced from thiol-ene combinations showed various intensities of coloration in reactions with DTNB. The most intense coloration was observed for thiol-acrylate combinations (P1, P2, P6, P7, P11, P12). In mixed mode photopolymerization between thiols and acrylates, acrylate monomers react in both homopolymerization and heteropolymerization with the thiol. This can be an evidence of higher consumption of –C=C during the mixed mode photopolymerization; consequently, more –SH could remain unreacted. In the products of pure thiol-ene polymerization reactions, less intense coloration was observed after the reaction with DTNB. Polymer nanoparticles from different monomer combinations disperse differently in PBS, and some materials tend to form aggregates. Quantitative analysis, therefore, cannot be carried out accurately. Nevertheless, these results confirm the abundance of reactive –SH groups within thiol-acrylate nanoparticles. The reactive –SH groups can be utilized for the post-polymerization functionalization via thiol-Michael ‘click’ reactions to conjugate the polymer nanoparticles with biomolecules for their future applications, for example, in biotechnology and biomedicine.

### 3.3. Size Distribution Studies

Polymer nanoparticles produced via aerosol photopolymerization are not monodisperse. Akgün et al. have studied the size distribution of the aerosol droplets and the polymer nanoparticles in an on-line measurement using scanning mobility particle sizer consisting of an electrostatic classifier and differential mobility analyzer. They have confirmed that the diameter of the majority of the polymer nanoparticles was below 300 nm [23]. Several factors affect the particle size and dispersity, while this paper only investigates the narrowing of the size distribution via decreasing the viscosity—i.e., increasing the ratio of solvent to monomers. An additional effect that must be considered is the evaporation of the relatively volatile solvent during the flight of the droplets until polymerization is completed. Significant changes to the size distribution of the nanoparticles are observed when the solvent amount exceeds the amount of monomers by 20 times (Figure 4). The size distribution narrows and shifts into the region of smaller nanoparticle size. These changes were confirmed from the analysis of SEM images by deducing histograms (Figure 4, right). All D-values were decreased from sample P1 to P1-S*20 (Figure 4, left): D50 from 398 to 247, D90 from 625 to 483, and D10 from 150 to 137. Low yield in highly diluted spray solutions is one of the drawbacks of using this method to narrow the size distribution of polymer nanoparticles. The 20-times increase in solvent (EtOH) amount leads to a three-fold decrease in yield. Throughout this paper, higher ratios of solvent to monomers were studied as well, decent mass yields (80 mg) can be achieved up to a 30:1 solvent to monomers ratio. The nanoparticles form aggregates in many solvents making use of other particle size analysis techniques unreliable. Dynamic light scattering (DLS) was one of the methods that were tested to acquire particle size distribution. The more stable dispersions of nanoparticles can be obtained in dimethylsulfoxide (DMSO). Dispersions of nanoparticles in DMSO can also be used for syringe filtration in order to remove bigger nanoparticles and keep the size distribution in a range between 50 and 500 nm.

## 4. Conclusions

This research demonstrates that polymer nanoparticles bearing active –SH groups can be produced from thiol and alkene monomers via aerosol photopolymerization. Combinations of trifunctional alkenes and trifunctional thiols, as well as trifunctional alkenes and tetrafunctional thiols, work best in an equal stoichiometry of functional groups to produce spherical individual cross-linked poly(thio-ether) nanoparticles. Several other monomer combinations—e.g., with difunctional alkene and trifunctional thiol (2FG–3FG)—work as well, but the produced nanoparticles are not as individual as nanoparticles from 3FG–3FG combinations. The poly(trithiol-TMPTA) nanoparticles contain no traces of unreacted -ene groups, leading to the conclusion that all TMPTA has reacted under cross-linking during photopolymerization. Poly(trithiol-TMPTA) nanoparticles were produced from mixed mode polymerization, where both homopolymerization of TMPTA and copolymerization of TMPTA and trithiol took place. Poly(trithiol-TATT) nanoparticles are a product of copolymerization of trithiol and TATT, due to the inability of TATT to produce nanoparticles via free radical aerosol photopolymerization. The products of the ‘true’ thiol-ene reaction (poly(trithiol-TATT)) do not have any thiol functionalities as seen on ATR-FTIR. These results also mark a good start in the study of mixed-mode polymerization reaction in aerosol thiol-ene photopolymerization systems, due to the high purity of polymer material collected in a dry state. The size distribution of nanoparticles was narrowed by increasing the solvent amount in the spray solution, simultaneously lowering its viscosity. The produced nanoparticles have active –SH groups that were traced via Elman’s test. The nanoparticles can further be adapted with biofunctionalization techniques for targeted drug delivery and other biomedical applications.

## Figures and Tables

**Figure 1 polymers-13-04363-f001:**
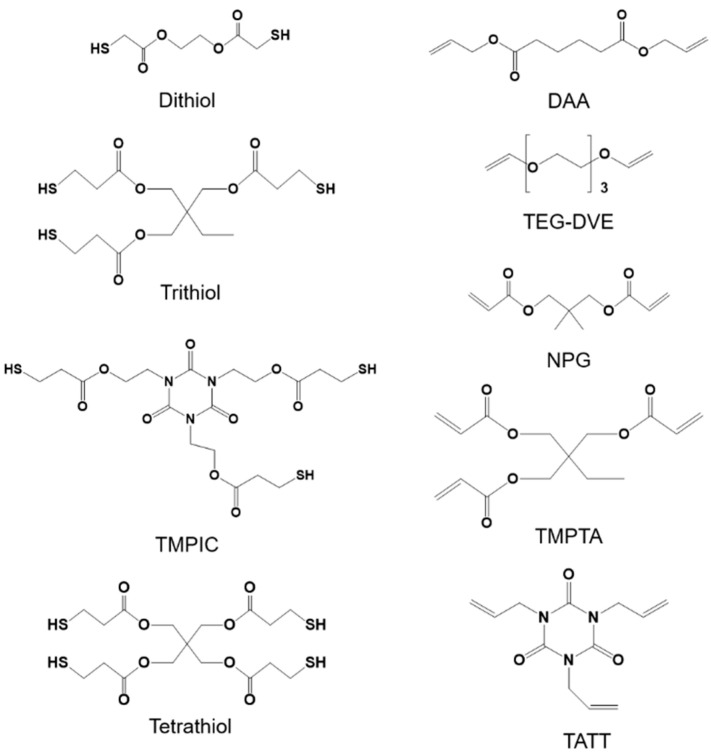
Thiol (**left**) and -ene (**right**) monomers used for the synthesis of polymer nanoparticles via APP.

**Figure 2 polymers-13-04363-f002:**
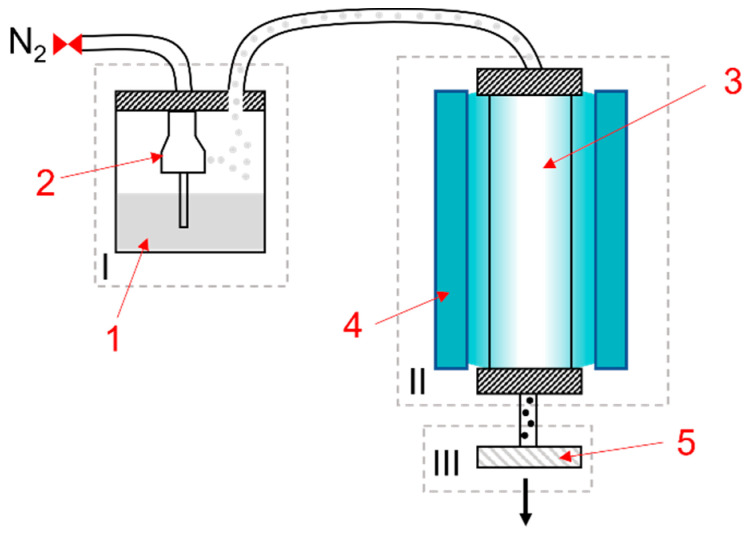
Aerosol photopolymerization setup: I—atomization unit; II—photoreaction unit; III—collection unit; (1) Spray solution, (2) Nozzle, (3) Reactor, (4) UV fluorescent devices, (5) particle collection (membrane filter housing).

**Figure 3 polymers-13-04363-f003:**
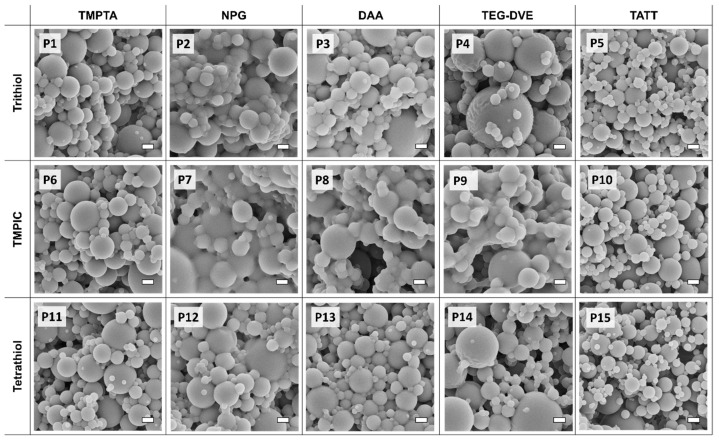
SEM images of polymer nanoparticles produced via aerosol photopolymerization from various thiol-ene monomer combinations. For simplicity, the nomenclature of thiol and alkene monomers is presented on the left-hand side and the top, respectively. The scalebar corresponds to 1 µm. The nomenclature agrees with the formulations listed in Table 1.

**Figure 4 polymers-13-04363-f004:**
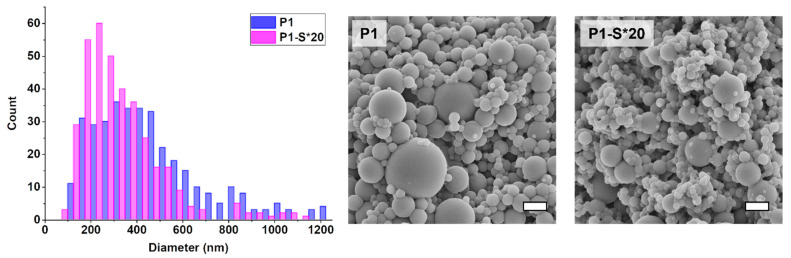
Size distribution histogram (**left**) as a function of particle diameter and particle count (graphed using image analysis of SEM micrographs) and corresponding SEM images of the polymer nanoparticles produced from solutions P1 and P1-S*20 (**right**). Scalebar—1 µm.

**Table 1 polymers-13-04363-t001:** Formulations of spray solutions with monomers and their quantities and the chosen solvent. The amount of the solvent is the same (10 g) in all samples, the amount of photoinitiator in all spray solutions is 1 wt% ratio to combined monomers. Equal stoichiometry (1:1 ratio) of functional groups (−SH for thiol and −C=C for double bonds) was considered in each spray solution.

Spray Solution	Thiol	Thiol Qtty (mM)	Alkene	Alkene Qtty (mM)	Functional Group Ratio	Solvent
P1	Trithiol	14.4	TMPTA	14.4	3−SH:3−C=C	EtOH
P2	Trithiol	13.9	NPG	20.9	2(3−SH):3(2−C=C)	EtOH
P3	Trithiol	13.5	DAA	20.3	2(3−SH):3(2−C=C)	EtOH
P4	Trithiol	14.2	TEG-DVE	21.3	2(3−SH):3(2−C=C)	EtOH
P5	Trithiol	15.4	TATT	15.4	3−SH:3−C=C	AcO
P6	TMPIC	12.1	TMPTA	12.1	3−SH:3−C=C	AcO
P7	TMPIC	11.8	NPG	17.8	2(3−SH):3(2−C=C)	AcO
P8	TMPIC	11.6	DAA	17.3	2(3−SH):3(2−C=C)	AcO
P9	TMPIC	12.1	TEG-DVE	18.1	2(3−SH):3(2−C=C)	AcO
P10	TMPIC	12.9	TATT	12.9	3−SH:3−C=C	MeCN
P11	Tetrathiol	11.3	TMPTA	15.1	3(4−SH):4(3−C=C)	MeCN
P12	Tetrathiol	10.9	NPG	21.9	4−SH:2(2−C=C)	MeCN
P13	Tetrathiol	10.6	DAA	21.3	4−SH:2(2−C=C)	MeCN
P14	Tetrathiol	11.2	TEG-DVE	22.3	4−SH:2(2−C=C)	MeCN
P15	Tetrathiol	12.2	TATT	16.2	3(4−SH):4(3−C=C)	MeCN

## Data Availability

Raw data of the SEM micrographs can be found in the following link: https://wetransfer.com/downloads/eaae5fb1ef3abae7cc2bd92021cd305f20211208192756/b621fe440e28e864e5839278d22d80b420211208192822/7e2faa (accessed on 20 November 2021).

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
