# Peer review of "Thiol-Functional Polymer Nanoparticles via Aerosol Photopolymerization"

_polymers, 2021, doi:10.3390/polym13244363_

Round 1
Reviewer 1 Report
The manuscript presented to me for revision „Thiol-functional polymer nanoparticles via aerosol photopolymerization” is very interesting and well written. The authors used the aerosol photopolymerization method to obtain polymer nanoparticles. They used monomers with different functionality (from 2 to 4) and different structures (alkyl, and with isocyanurate ring) to check how these parameters will influence the size and its distribution of synthesized polymer droplets. The study is well planned, however, more information about conversion degree or kinetics of polymerization of the investigated systems could be added. Moreover, the stoichiometric ratio of functional groups SH:CC is very important in thiol-ene polymerization. The authors used various ratios of SH:CC groups, so this information should be added to the manuscript, e.g, in table 1. This will make understanding of obtained results easier. Additionally, can the authors give more information about the results of Ellman’s test? There is information about various intensities of coloration in reactions with DNTB. But authors didn’t associate these results with the composition (monomers structure) of the used systems. Is there any correlation?
Author Response
Dear reviewer,
Thank you for taking time to assess my manuscript. I have adressed all concerns you had reagarding my work and here are a few answers to your specific comments, which improved my manuscript:
- conversion degree and kinetics of polymerization: we could not determine conversion using NMR, because our monomers are liquid and polymer nanoparticles are solid and due to cross-linking are insoluble in NMR solvents. Calculating conversion from liquid to solid NMR will not give reliable results. Although, full consumption of double bonds in Trithiol-TMPTA combination was stated in the paper (Supporting Information I, page 1), meaning 100% conversion of double bonds was achieved. Calculation of the conversion was attempted using FTIR (for all combinations) and Raman (for some combinations), but none of the methods gave reliable results that can be correlated with monomer and polymer composition (chain-, thiol-ene, mixed-mode polymerization). For this investigation it would be essential to select monomers to be compatible with the characterization methods, especially in the polymerized state. We presented some information on conversion of double bonds and -SH groups in Supporting information.
- stoichiometry of functional groups: I added the FG ratio to the Table 1 (chapter 2, page 5) and a short explanation in the text. We always used 1 to 1 ratio of SH and CC groups.
- Ellman's test: there is indeed a correlation between the choice of monomers and the presence of -SH groups on the surface of nanoparticles. In case of a pure thiol-ene reaction, we usually observed lower intensity of coloration after Ellman's reaction. For mixed mode polymerization (thiol-acrylate combinations) we observed very intense coloration. I have added some text to the section of concern, please have a look at the revised version, specifically Chapter 3, pages 7 and 8. It is quite difficult to perform quantitative analysis of results of Ellman's experiment in our heterogeneous systems. Some polymer nanoparticles form aggregates in PBS, whereas others form colloidally stable dispersions.
Kind regards,
Narmin Suvarli
Reviewer 2 Report
In this paper the authors investigate aerosol photopolymeryzation of thiol-ene formulations. The work is well performed and well written, I have only a small doubt regarding section 3.2: the authors refer to a change of color, even with various intensities. Hoverver there is not evidence of that (UV vis or at least pictures) so I believe that this section should be improved.
Then I believe it can be published.
Author Response
Dear reviewer,
Thank you for taking time to assess my manuscript. I have adressed all concerns you had reagarding my work and here is the answer to your specific comment, which improved my manuscript:
- Ellman's test: there is indeed a correlation between the choice of monomers and the presence of -SH groups on the surface of nanoparticles. In case of a pure thiol-ene reaction, we usually observed lower intensity of coloration after Ellman's reaction. For mixed mode polymerization (thiol-acrylate combinations) we observed very intense coloration. I have added some text to the section of concern, please have a look at the revised version, specifically Chapter 3, pages 7 and 8. It is quite difficult to perform quantitative analysis of results of Ellman's experiment in our heterogeneous systems. Some polymer nanoparticles form aggregates in PBS, whereas others form colloidally stable dispersions.
Kind regards,
Narmin